# A Proposed Non-Destructive Method Based on Sphere Launching and Piezoelectric Diaphragm

**DOI:** 10.3390/s24185874

**Published:** 2024-09-10

**Authors:** Cristiano Soares Junior, Paulo Roberto Aguiar, Doriana M. D’Addona, Pedro Oliveira Conceição Junior, Reinaldo Götz Oliveira Junior

**Affiliations:** 1Department of Electrical Engineering, São Paulo State University (UNESP), Av. Eng. Luiz Edmundo C. Coube 14-01, Bauru 17033-360, Brazil; cristiano.soares@unesp.br (C.S.J.); paulo.aguiar@unesp.br (P.R.A.); 2Department of Chemical, Materials and Industrial Production Engineering, University of Naples Federico II, 80138 Napoli, Italy; 3Department of Electrical and Computer Engineering, University of São Paulo (USP), Av. Trabalhador São-Carlense, 400, São Carlos 13566-590, Brazil; pedro.oliveiracjr@usp.br; 4Federal Institute of Education, Science and Technology of Mato Grosso do Sul (IFSM), R. Ângelo Melão, 790, Três Lagoas 79641-162, Brazil; reinaldo.gotz@usp.br

**Keywords:** non-destructive test, piezoelectric transducer, SHM, CFRP, PLB

## Abstract

This work presents the study of a reproducible acoustic emission method based on the launching of a metallic sphere and low-cost piezoelectric diaphragm. For this purpose, tests were first conducted on a carbon fiber-reinforced polymer structure, and then on an aluminum structure for comparative analysis. The pencil-lead break (PLB) tests were also conducted for comparisons with the proposed method. Different launching heights and elastic deformations of the structures were investigated. The results show higher repeatability for the sphere impact method, as the PLB is more affected by human inaccuracy, and it was also effective in damage detection.

## 1. Introduction

Carbon fiber-reinforced polymer (CFRP) has become a widely used material in modern manufacturing due to its application in vital parts to structural safety in durable consumer goods [1]. High stiffness, low weight, and increased fatigue strength are the main reasons that make CFRP so attractive to the industry. However, CFRP is susceptible to damage during machining, such as fiber cracks, matrix defects (gap, porosity), and delamination [2]. Delamination is one of the most common failure forms of composites and occurs because of continuous stress and pressure, which causes layer deformation [3]. It is responsible for drastically reducing the mechanical resistance of this material [4] and has been reported in reviews of drilling [5] and milling [6] operations of this composite.

To monitor the integrity of CFRP structures, several monitoring methods have been proposed during the manufacturing stages. Destructive methods, which are less commonly preferred because the specimen under test undergoes continuous load application until it eventually fails, are sometimes necessary for comprehensive analysis. Bending tests were performed for research purposes with the aim of using CFRP for structural reinforcement [7]. Tensile strength tests were conducted to perform a failure analysis of CFRP anchors [8]. The effect of elevated temperatures on the microhardness of CFRP was investigated using the Vickers hardness test method, which is widely used for quality control [9]. On the other hand, non-destructive testing (NDT) plays a key role in monitoring manufacturing processes. Direct monitoring techniques can be employed to verify structural integrity, such as visual inspection [10], electrical resistance [11], and infrared thermography with a laser beam [12]. However, indirect monitoring techniques are preferred today because they are helpful in the online measurement of the process responses [13]. These methods are commonly used for structural health monitoring (SHM) in several knowledge areas, such as bridges in civil engineering [14], aircrafts in aerospace engineering [15], and metallic structures in mechanical engineering [16]. The SHM aims at strategies to identify damage in structures by performing periodic measurements to determine the degradation conditions of components [15].

The different SHM methods applied in CFRP structures can be classified as active and passive. In passive detection, it is usual to use non-destructive testing based on acoustic emission (AE) techniques since failures in composite materials produce frequencies that can be detected using AE sensors [17]. Acoustic emission can be defined as transient elastic waves naturally generated within a medium due to a rapid release of energy resulting from microstructural changes, such as cracking or plastic deformation. These waves can be detected and converted to a voltage signal by an AE sensor attached to the structure surface being monitored [18]. It is a continuous monitoring technique rather than a tool for non-destructive inspections, and the structure must be loaded or eventually damaged to collect acoustic emissions [19].

Many studies have been conducted to detect composite delamination based on the passive method of AE and signal processing, such as using counts metric [20], cumulative energy [21], frequency analysis based on fast Fourier transform (FFT) [22], and convolutional neural networks (CNN) [23]. Active methods, on the other hand, use piezoelectric transducers that inject controlled signals into the structure, such as the electromechanical impedance method [24], Lamb waves [25], and guided waves [26].

According to [27], SHM techniques using ultrasonic-guided waves can be divided into passive and active monitoring. Passive monitoring refers to the monitoring of guided ultrasonic wave signals generated by an emergency, such as an external impact, where the signals can be captured and identified by a sensor. Active monitoring refers to a guided Lamb wave signal generated by excitation transducers and wave scattering caused by damage. The work of [28] states that guided wave (GW) testing has emerged as a very prominent option among active schemes. It can offer an effective method to estimate the location, severity, and type of damage, and it is a well-established practice in the non-destructive evaluation and testing (NDE/NDT) industry. In addition, according to the authors, to distinguish between damage and structural features, prior information is required about the structure in its undamaged state. This is typically in the form of a baseline signal obtained for the “healthy state” to use as a reference for comparison with the test case. 

Thus, this work investigates a novel NDT approach for structural health monitoring of CFRP using an acoustic wave generation method based on launching a metallic sphere on the surface of a CFRP plate. It is a passive guided wave technique according to the definition of [27], where the impact of the sphere onto the structure surface under test acts as a source of acoustic waves to interrogate the structure. A piezoelectric diaphragm is placed on the opposite side of the structure to collect the signals for analysis. The pencil-lead break (PLB) method was also used as a reproductive acoustic wave-generating method to compare the results with the proposed method. The PLB method was used for comparison because it is a well-known and established method in the literature for characterizing and evaluating the frequency response of piezoelectric transducers and AE sensors, as used by [29].

It is worth mentioning that the proposed method in this work has not yet been reported in the literature as presented herein, and it is highlighted as an alternative non-destructive testing methodology for CFRP structure evaluation. Also, this work explores the features of the signals collected through signal processing, as measurements of the acoustic emission are impacted by attenuation, dispersion, and reflection during the time it propagates from source to sensor, and therefore the waveforms, amplitudes, and frequency content of a wave packet differ [30].

It is also noteworthy that some research works on dropping a ball on the structure surface as a source of AE can be found, but with different purposes compared to the methodology proposed herein. In the work of [31], a technique to estimate the seismic moment of acoustic emissions and other extremely small seismic events is reported, which calibrates the recording system as a whole and uses a ball impact as a reference source. The paper of [32] investigated the impact behavior and damage mechanism of a carbon-Dyneema hybrid fabric reinforced composite through drop-weight and steel ball impact tests. The experimental results show that the impact resistance of the hybrid composites was improved when benchmarked with carbon fiber composites. The authors of [33] studied the damping properties of graphene-modified polyurethane for three different frequency ranges by using a drop ball test. In the concluding remarks, the authors state that in the drop ball test, the amount of voltage recorded by the piezoelectric sensor is very sensitive to the experimental setup, test specimen preparation, and surface condition of the specimen. Lastly, the work of [34] investigates the influences of backing layers on the electroacoustic properties of the AE sensors based on the pencil-lead break test and simulated noise experiments using steel ball drop. According to the authors, a steel ball of 11 mm in diameter was freely dropped from a height of 20 cm above the center of the aluminum plate, and the stress wave produced by the impact of the steel ball on the aluminum plate was simultaneously received by the acoustic emission sensors. When the steel ball hit the aluminum plate, a brush was simultaneously used to rub the aluminum plate to produce environmental noise.

In this context, the main contributions of the present work and features that make it innovative are as follows:A method for characterization and frequency response evaluation of piezoelectric transducers as an alternative to the PLB method for CFRP structures;A new approach for structural damage evaluation in CFRP as a potential acoustic technology for various NDT applications, contributing to the further knowledge of damage mechanisms and acoustic propagation in CFRP;A simple and low-cost methodology using a piezoelectric transducer and inexpensive apparatus, which enables flexibility compared to other SHM approaches;The non-destructive evaluation method can monitor the structure through sphere launching without causing any permanent damage to the CFRP specimen.

To validate the effectiveness of the proposed method, two different kinds of experimental tests based on sphere launching were performed on a CFRP structure. The first test consists of performing consecutive launches at a specific point on the plate surface, at different heights between each test. The second test consists of performing launches on the plate surface at the same point, simulating structural damage by attaching metallic nuts to the plate surface under study. Similar tests were conducted on the same-sized aluminum plate for comparative purposes to evaluate the proposed method for different materials, expanding its contributions. The PLB method was also employed in both materials to assess the structure conditions for comparative analysis.

## 2. State of the Art

### 2.1. CFRP Composites

CFRP composite materials have been widely used in engineering to manufacture critical components because of their high specific stiffness, high specific strength, superior corrosion resistance, and excellent thermal stability [35]. In the aircraft industry, an increasing number of components are made from CFRP aiming to reduce the weight of the structure, improve fuel efficiency, reduce emissions, and enhance load-bearing capacity [2]. In automotive engineering, CFRP is a suitable material, despite its high cost. It can reduce weight, integrate parts, improve crashworthiness, increase durability and toughness, and provide aesthetic appeal [36].

Researchers have investigated the effects of CFRP machining parameters on several failures (delamination, matrix defects, fiber cracks, impurities, interface cracks, etc.) by traditional methods (drilling, milling, turning, etc.) and non-traditional methods (water jet machining, abrasive water jet machining, laser machining, electrical discharge machining, ultrasonic machining, electrochemical machining, etc.) [2].

Table 1 compares some features between CFRC and aluminum, another widely used material in the industry. Aluminum is presented here because it was used as a comparative material in the experiment, as described in Section 3.

Based on their respective densities, CFRC is found to be significantly lighter than aluminum. Aluminum is a ductile material that can be molded and extruded in several ways. Both CFRC and aluminum are highly resistant to corrosion. Aluminum can provide good mechanical resistance when combined with other materials, forming alloys [37]. However, aluminum still has less resistance and stiffness compared to CFRC. The cost of manufacturing aluminum is usually lower than that of CFRC because it employs well-known methods, such as casting, machining, and forging, while CFRC requires more complex processes, such as lamination and autoclave curing [38].

### 2.2. The PLB Method

The PLB is a well-established method of using pencil-lead breaks as AE sources. It consists of breaking a lead at a specified angle to the material surface under study. The lead break releases energy in the form of an elastic wave that propagates through the entire structure, making it a simple and reliable way to create a broadband signal to study the wave propagation and the acoustic sensor response in the host structure. However, conducting PLB tests with a high degree of repeatability can be challenging in practice [39].

Several recent studies have used the PLB method to generate acoustic waves. In a study by Yao et al., they decomposed AE excited by PLB on a thin isotropic plate through modal decomposition using a multi-element piezoelectric sensor array. According to the study, the direction of the PLB orientation has an impact on AE modal decomposition [40]. In a study by Ghadarah and Ayre, the PLB method showed that an AE transducer fully embedded in a glass fiber composite presents lower sensitivity than an AE transducer surface embedded [41]. Mahajan and Banerjee conducted a study in which they located an AE source, which was simulated using PLB as a source, based on a deep learning algorithm [42].

Almeida et al. investigated the piezoelectric transducers employed in the PLB method and assessed the sensitivity of piezoelectric transducers for damage detection [43]. Piezoelectric diaphragms are commonly used in audio signaling devices because of their low cost and simple structure. The diaphragm consists of a piezoelectric ceramic disc attached to a brass plate, and it is easily applied to surfaces [44]. Piezoelectric diaphragms have been used in several monitoring applications, such as the grinding process [45], water flow rate in a chamber [46], and partial discharge detection [47].

### 2.3. Damage Indices

Previous studies on SHM have used statistical parameters to characterize damage, such as root mean square deviation (RMSD) and the correlation coefficient deviation (CCDM) [45]. Transducers are connected to a data acquisition (DAQ) system. This system collects mechanical sound waves that propagate through the structure and transfer them to a computer for digital processing. Regions of interest can be selected in the time domain, and frequency bands can be filtered using frequency spectrum analysis. To search for variations in the frequency content that are correlated to the structural conditions, the RMSD and CCDM damage indices can be calculated using Equation (1) and Equation (2), respectively [44].
(1)$$RMSD=\sum_{{k=\;\omega }_{I}}^{\omega_{F}}\sqrt{\frac{{\left[X_{D}\left(k\right)-X_{H}\left(k\right)\right]}^{2}}{X_{H}^{2}(k)}}$$
(2)$$CCDM=1-\frac{\sum_{\omega_{I}}^{\omega_{F}}\left[X_{H}\left(k\right)-{\bar{X}}_{H}\right]\left[X_{D}\left(k\right)-{\bar{X}}_{D}\right]\;}{\sqrt{\sum_{\omega I}^{\omega F}{\left[X_{H}\left(k\right)-{\bar{X}}_{H}\right]}^{2}}\sqrt{\sum_{\omega I}^{\omega F}{\left[X_{D}\left(k\right)-{\bar{X}}_{D}\right]}^{2}}}$$

In Equations (1) and (2), $X_{H}\left(k\right)$ and $X_{D}\left(k\right)$ are the spectral frequency signatures for the healthy (H) structure at a frequency k and the damaged (D) structure, respectively, and they are computed in the range of frequency selected between ω_I_ and ω_F_. In Equation (2), ${\bar{X}}_{H}$ and ${\bar{X}}_{D}$ are the mean values of the signatures.

## 3. Material and Methods

The method proposed in this study is based on the propagation of acoustic emission waves, which are generated by the impact of the sphere on the structure. The waves change based on material properties and existing damages. To validate the effectiveness of the proposed method, two different kinds of experimental tests based on sphere launching were performed on a CFRP structure. The first test consists of performing consecutive launches at a specific point on the plate surface, at different heights between each test. The second test consists of performing launches on the plate surface at the same point, simulating structural damage by attaching metallic nuts to the plate surface under study. Similar tests were conducted on the same-sized aluminum plate for comparative purposes to evaluate the proposed method for different materials, expanding its contributions. The PLB method was also employed in both materials to assess the structure conditions for comparative analysis.

### 3.1. Experimental Setup

A workbench schematic diagram of the launching sphere method is presented in Figure 1. One notes that the sphere’s collision with the plate generates acoustic waves on the material. The signals are collected by the piezoelectric transducer, which is connected to an oscilloscope, and digitally processed on MATLAB^®^ software (Natick, MA, USA).

The experimental workbench is shown in Figure 2. To conduct the experiments, a 19.9 mm diameter piezoelectric diaphragm was attached at 25 mm from the edge of a CFRP plate with 448 × 106.3 × 5 mm dimensions and 391.46 ± 1.03 g mass. The CFRP plate was the same one used in [48], the EXP-C, which has 24 layers of 0/90° bidirectional plain-weave fiber fabric, Hexcel^TM^ (SJ Campos, Brazil), as illustrated in (2) in Figure 2. The point where the metallic nuts were attached for damage simulation was 105 mm from the piezoelectric diaphragm, as in (4) in Figure 2. The hexagonal metallic nuts have dimensions of 6.2 mm diameter and a mass of 0.54 ± 0.02 g, as in (8) in Figure 2. Underneath the CFRP plate, a foam sheet of the same dimensions was arranged to avoid any unwanted external vibration, as shown in (5) in Figure 2. A thin layer of ethyl cyanoacrylate-based glue was used to fix the piezoelectric transducer. The steel sphere used in the launches, which comes from a bearing, has a 6.3 mm diameter and 1.046 ± 0.01 g mass and is illustrated in (9) in Figure 2. The selection of this sphere was empirical, and the choice was based on its availability, reduced size and weight, and the reduced potential to cause damage on the surfaces of both materials when launched at a distance of up to 500 mm. To suspend the sphere over the plate and perform the launches, it was required to use a device consisting of a wooden base, a metal rail, and an electromagnet attached to the rail, as illustrated in (6) in Figure 2. The launch base of the sphere, equipped with an electromagnet, is coupled to the rail of 540 mm in height to allow the adjustment of the launch height. Once the height is adjusted, a permanent magnet fixes the launch base at the desired height. To control the launch height, five equal aluminum blocks 38 mm high each were used as templates (models), as shown in (10) in Figure 2. An adjustable voltage source is used to power the electromagnet, as in (7) in Figure 2. The acquired signals were collected and stored by an oscilloscope model DL850 from Yokogawa^®^ (Musashino, Japan), as in (1) in Figure 2, at a sampling frequency of 10 MHz.

After the acquisition, all data were transferred to a computer, where the signals were digitally processed using MATLAB^®^ software. The digital signal processing procedure initially consisted of segmenting each signal to extract only the signal related to the first touch of the sphere on the surface of the structure, eliminating the parts of the signal regarding the sphere rebounds. The frequency spectrum is computed, analyzed, and compared to search for variations in the frequency content correlated to the structural conditions. Subsequently, RMSD and CCDM damage indices are computed.

To assure the effectiveness of the proposed method and allow comparisons, the tests were repeated under the same conditions for an aluminum structure with the same dimensions as the CFRP plate, but with 643.4 ± 0.2 g mass, as illustrated in (3) in Figure 2. The piezoelectric diaphragm was also fixed at 25 mm from the edge of the aluminum plate.

### 3.2. PLB Tests

The PLB tests were conducted by breaking a 0.5 mm thick lead at a 45° angle to the CFRP plate surface at two positions, one at 70 mm and the other at 210 mm from the diaphragm. Five repetitions were performed in each defined position, and then the correlation coefficient was calculated for each position, as seen in [49], to evaluate the repeatability and feasibility of the repetitions.

Similarly, to the CFRP structure, PLB tests were conducted on an aluminum plate surface. For this purpose, three repetitions were performed at 210 mm from the piezoelectric diaphragm. The distance was chosen based on the results from the CFRP tests and the correlation coefficients were calculated to evaluate the repeatability between the tests.

### 3.3. Plastic Deformation Evaluation in Sphere Launch Tests

Preceding the sphere launch tests, tests were conducted to evaluate CFRP and aluminum plastic deformation caused by the metal sphere impact on the structure’s surface. For this purpose, a series of launches were performed at different heights. Through a digital microscope from Inskam, 30 W, and at 1000× magnification, it was possible to obtain detailed images of the structures’ conditions before and after the launches.

For the CFRP tests, ten consecutive launches were performed at the same point of the plate surface, using a 6.3 mm diameter sphere to evaluate whether there would be an onset of marks or cracks on the surface. The tests were conducted at three launch heights (114 mm, 152 mm, and 190 mm).

For the aluminum tests, two metal spheres were used. One sphere has 3.9 mm in diameter and 1.046 ± 0.006 g mass. The other one was the same 6.3 mm diameter sphere used previously. The aluminum tests consisted of launching each sphere from three different heights (114 mm, 152 mm, and 190 mm). The reason for using two spheres for the aluminum was due to marks visually observed and caused by the impact of the 6.3 mm sphere in the preliminary tests.

### 3.4. CFRP Launch Tests for Height Variation Study

Each test consisted of launching a 6.3 mm diameter and 1.046 ± 0.006 g metal sphere on a CFRP plate surface at 210 mm from the piezoelectric transducer, based on the results of Section 3.2. Five aluminum blocks, 38 mm long each, were used as templates to ensure the exact launch height for each test. Thus, initially, there were five launching heights: 38 mm, 76 mm, 114 mm, 152 mm, and 190 mm. A sixth height of 490 mm was subsequently defined based on the maximum size of the metallic rail. Therefore, for each of the six defined heights, five launches were performed on the CFRP surface. Then, the correlation coefficient was calculated to verify the similarity between the launches’ repetitions. It is worth mentioning that all tests described in this section were conducted at a constant temperature of 25 ± 2 °C. The signals were segmented to extract only the signal generated from the first touch of the sphere on the CFRP surface, thus removing the signals obtained from the rebounds. The frequency spectra were computed and compared to verify variations as the launch height was changed. An average of the five spectra obtained for each height was calculated to simplify the visualization and analysis.

### 3.5. Aluminum Launch Tests to Verify the Effect of Sphere Damage

Preliminary tests have shown that the sphere launches on the aluminum have generated visible marks (possible damage), which could compromise the repeatability of the tests for this material. Thus, launch tests were performed on aluminum to verify this fact, which consisted of launching the 6.3 mm diameter metallic sphere seven consecutive times at a 114 mm height. The sphere impact location on the aluminum surface was defined at 210 mm from the piezoelectric diaphragm based on the results in Section 3.2. The height of 114 mm was chosen based on the highest correlation and lowest standard deviation obtained for the launches in CFRP. The tests were conducted at a constant temperature of 25 ± 2 °C. Then, the correlation coefficient between the first launch (initial mark of the sphere) and the following launches was calculated. Finally, the signal spectra were computed and analyzed.

### 3.6. SHM Tests

Considering the studies previously conducted, sphere launch tests were performed on both CFRP and aluminum plates, simulating structural damage by adding mass to their surfaces. For this purpose, five metallic nuts were gradually added to the surface of both plates by an ethyl cyanoacrylate-based adhesive. Before the mass addition, five launches were performed to characterize the baseline for both plates. After each mass addition, five more launches were performed on the same point at 210 mm from the piezoelectric transducer fixed at 25 mm from the edge of both plates. The metallic nuts were attached and stacked one by one at 105 mm from the transducer. Similarly, the PLB tests were conducted on the same structures of CFRP and aluminum by breaking the pencil lead at the same point where the sphere was launched. Five repetitions were also performed for the baseline and five metallic nuts of the same specifications were gradually added to the surfaces. The PLB tests in this step of methodology were meant to be compared to the launch tests regarding the detection of damage to the structures.

## 4. Results and Discussion

This section will present all the results and discussions from the experiments described in the previous section.

### 4.1. Results of SHM Tests

Figure 3 shows the microscopic images obtained for CFRP before (a) and after (b) the sphere impact for three launching heights. It can be observed in the circled areas where the impacts occurred that there is no apparent damage at the tested heights.

The microscopic images obtained for the aluminum are shown in Figure 4. In the images, the marks resulting from the sphere’s impact on the aluminum plate surface after the first launch are evident. The 3.9 mm sphere caused circular damage of about 0.5 mm diameter, while the 6.3 mm sphere caused similar damage of about 0.8 mm diameter. However, it is seen that the damage caused by the 3.9 mm sphere is deeper. The smaller marks around the impact area are due to the spheres rebounding off the plate surface.

### 4.2. Results from PLB Tests on CFRP

Table 2 shows the correlation coefficients between the repetitions obtained for the time domain signals from the PLB tests on the CFRP. It is observed that the coefficient for both positions is close to 93% with low standard deviations between the repetitions, which demonstrates good repeatability between the tests. Position 2 was chosen in this work due to the greater easiness of using the sphere launching device. Figure 5 shows the raw signal of repetition 1.

Figure 6 shows the average frequency spectra. The amplitudes vary based on the test location, alternating between the lead break positions along the spectrum. In certain ranges, the amplitudes are higher for position 1 (70 mm), as demonstrated by the amplification of the 0 to 3 kHz band in Figure 6. Spectral activity is observable up to about 25 kHz, especially in the range of 0 to 10 kHz, where amplitudes are higher. The results showed that the piezoelectric transducer and CFRP assembly are sensitive to the impact generated by the lead breaking on the plate surface. It is worth mentioning that CFRP is a heterogeneous and anisotropic material, with much lower density than metals, providing that the dispersive and dominant flexural out-of-plane Lamb waves easily travel on its surface. Furthermore, acoustic waves propagating in a fiber-reinforced composite are subject to scattering and attenuation, even though the whole system is perfectly elastic. This is due to the multiple dispersion of incident waves through the randomly distributed composite fibers [50,51].

### 4.3. Results from PLB Tests on Aluminum

The correlation coefficients between the PLB tests (repetitions) on aluminum are shown in Table 3. It is observed that the values are extremely high, demonstrating great repeatability for this material.

The spectra for each PLB test are shown in Figure 7. It is possible to notice wider frequency responses for aluminum when compared to CFRP but with lower amplitudes.

The aluminum density is higher than CFRP, and therefore the out-of-plane dislocations of flexural mode are smaller at the surface, as those waves are dispersive with velocities changing with frequency, resulting in lower amplitudes of the acoustic signal [51]. It is worth mentioning that the tests (lead breaking) on both plates were conducted at the same distance from the transducer, thus allowing a comparison of acoustic wave propagation in these materials.

### 4.4. Results from CFRP Launch Tests

Table 4 shows the correlation coefficient values between the repetitions for each launch test height for CFRP.

The correlation coefficients were calculated to evaluate the repeatability of the proposed method for acoustic wave generation and to define the best launch height. The mean and standard deviation of all possible combinations were computed for the five launches performed at each height. Notably, the correlation coefficient for all launch heights was greater than 99%, which shows the high repeatability of the method. However, the highest correlation value and the lowest standard deviation were obtained for the 114 mm height. It is worth mentioning that this height allows quick and easy handling of the metallic sphere without taking the risk of accidentally moving the plate or the electromagnet between launches.

The time domain raw signals were segmented to extract only the first sphere touch on the CFRP surface. The analysis of the raw signal obtained from the sphere launch tests on CFRP reveals that the highest amplitudes presented are obtained from higher launch heights, as seen in Figure 8.

Figure 9 presents the frequency spectra from the raw acoustic signals acquired from the sphere launch tests performed on the CFRP structure. As can be seen, the signal amplitude varies with the launch height, as was observed in the raw signal. The frequency range between 13 kHz and 14.8 kHz, illustrated in Figure 9, shows in detail the amplitude variation caused by the launch height variation, i.e., the higher the launch height, the higher the amplitudes obtained. This behavior is almost repeated across the entire spectrum, where spectral activity is detected up to about 25 kHz.

### 4.5. Results from Aluminum Launch Tests

From the raw signals of the seven launches on the aluminum, the correlation coefficient was computed between the first launch (baseline) and the other six launches. Table 5 shows only three of the computed coefficients, with these values being higher than 99%, demonstrating the deformation caused by the contact between the sphere and the plate surface, as observed in Section 4.1, Figure 4, does not affect the acquired signals, even if the tests are performed several times at the same point. This confirms the high similarity between the signals from several launches performed at the same surface position.

Figure 10 illustrates the frequency spectra obtained from the sphere launch tests on the aluminum surface. Similar to the PLB test, it is observed that the spectra show a wider frequency band and smaller amplitudes when compared to the amplitudes obtained from the CFRP. When analyzing a smaller frequency band (up to 4 kHz), it is possible to observe an almost complete overlap of the signals, indicating the high repeatability between the tests.

### 4.6. Results from NDT Tests

The frequency spectrum was computed, and an average of the five spectra obtained for each plate condition was calculated to simplify the analysis, as shown in Figure 11.

Through visual analysis of the CFRP spectrum, it is possible to notice amplitude variations in the signals between the structure conditions, as well as their frequency content. As a result of the composite materials’ attenuating conditions, it is possible to observe spectral activity up to approximately 30 kHz. But only through a detailed analysis of the entire transient signal, it was possible to select the 1 kHz to 9 kHz frequency band, which shows a more pronounced amplitude and frequency shifts over the whole signal. This frequency band clearly shows a signal variation due to the plate mass variation, evidencing the CFRP and piezoelectric diaphragm assembly sensitivity to mass variation and the impact caused on the structure surface by the proposed acoustic wave generation method. The frequency spectra for aluminum are shown in Figure 12.

The behavior observed in aluminum is very similar to that seen in CFRP. Still, the aluminum frequency ranges are much wider due to the material’s properties, reaching up to approximately 100 kHz. The amplitude variation and signal frequency shift concerning the aluminum plate mass are significant, thus evidencing the applicability of the acoustic wave generation method proposed for aluminum plates and its sensitivity to the mass variation. By analyzing the spectrum along the entire transient signal, it was possible to find the frequency band between 8 kHz and 16 kHz, where the amplitude and frequency content variation are more pronounced.

To evaluate and quantify the signal variations due to structure mass variation, the CCDM and RMSD indices were computed. Both metrics were applied for the 8 kHz to 16 kHz frequency band for both materials. Figure 13 shows the CCDM index for the CFRP structure.

It can be observed in Figure 13 that the CCDM index increases with the addition of metallic nuts to the plate surface, i.e., the increase in the CFRP plate mass causes a frequency shift between the spectra. Therefore, the CCDM index applied in the selected frequency band is able to identify and quantify the CFRP plate mass variation.

The CCDM was computed for aluminum, as shown in Figure 14, where a similar behavior to that reported for CFRP is observed. The progressive increase in the CCDM index indicates that the increasing structure mass leads to a shift in the frequency component of these signals, demonstrating that the used metric is sensitive to mass variation and, therefore, able to identify the simulated damage on the aluminum structure, as in the CFRP.

Figure 15 shows the RMSD index for the tests performed on the CFRP structure, where it is possible to observe an increasing RMSD index as the structure mass increases, indicating that the amplitude of the signals, acquired from the CFRP tests, increases in the selected frequency band as the structure mass increases.

Similarly, the RMSD index for the aluminum test signals is shown in Figure 16, i.e., there is a gradual index increase as the masses are added to the structure.

### 4.7. Results from NDT PLB Tests

The average spectra of PLB tests for the CFRP and aluminum plates are shown in Figure 17 and Figure 18, respectively. Similar frequency content and amplitude behavior to the launch tests can be observed in these figures, but the amplitudes are significantly smaller due to the very low mechanical energy impressed by the pencil-lead break compared to the impact of the sphere on the surface. On the other hand, a closer look along the spectra reveals different frequency bands with more pronounced amplitude variation and shifts for both PLB and launch methods. This difference may be explained by the nature of each method; that is, the PLB method imposes lower mechanical energy when the pencil lead is broken when compared to the impact of the sphere. As a result, the acoustic waves released in the structure will not be able to propagate in the same way as in the sphere impact, which will have different levels of transmission, reflection, attenuation, and interferences throughout the structure. A frequency band from 8.2 kHz to 8.8 kHz and another from 20 kHz to 24 kHz were selected for CFRP and aluminum plates, respectively.

To show the significant difference between the signal energy between both PLB and launch tests, the signal energy from the launch tests as well as from the PLB tests were computed, considering a sliding window of 0.5 milliseconds and the signal average of the repetitions for each condition. Figure 19 shows the signal energy for baseline and damage 1 for PLB and launch tests, as the other conditions show similar results. It can be initially observed that the signal energy of the PLB tests is significantly smaller than that of the signal energy of the launch tests for both conditions. The area under the curve (AUC) of each condition was computed and is shown in the figure. To better illustrate these figures quantitatively, the energy ratio between the launch and PLB tests was computed for both conditions, which is equal to 38.98 for baseline and 39.30 for damage 1, demonstrating a significantly greater signal energy from the launch tests.

The CCDM and RMSD indices computed in the band of 20 kHz to 24 kHz for the aluminum plate are shown in Figure 20 and Figure 21, respectively. As these metrics imply, the sequential increase in these indices is a measure of the progressive increase in signal amplitudes (RMSD) and shifts (CCDM) of each structure-condition spectrum with regard to the baseline.

Therefore, these results show a very good correlation between the indices and the mass variation. The results for the selected frequency band of 8.2 kHz to 8.8 kHz for the CFRP plate show a good correlation to the mass variation as well, but they are not presented herein to avoid repetitiveness. It is worth mentioning that other frequency bands were also found and presented good indices results.

### 4.8. Coherence between Launch and PLB Tests

To measure how well the signals from the launch tests correspond to the signals from the PLB tests at each frequency, the magnitude-squared coherence (MSC) was computed, as applied in the work of [52]. The MSC of each repetition was computed for each structure condition and material, and then an average was obtained. The function named coherence available in MATLAB was employed with a hamming window of 218 lengths and an overlap of 50%. Figure 22 and Figure 23 show the MSC between the launch and PLB tests for CFRP and aluminum, respectively.

It can be observed in Figure 21 that the average of the MSC for each CFRP structure condition clearly shows a good correlation (mean value higher than 0.8) between the launch and PLB signals up to 31 kHz. There is, however, a degradation of signals’ correspondence from this frequency, with the worst MSC of about 0.17 in the frequency band from 44 kHz to 48 kHz. In the case of the aluminum structure, as shown in Figure 23, the MSC average for each structure condition also presents a good correlation (mean value higher than 0.7) up to nearly 86 kHz, with a degradation of the signals’ correlation from this frequency, reaching the MSC of 0.21 in the frequency band from 86 kHz to 100 kHz. It can therefore be inferred that both launch and PLB methods have a good correlation in most of the useful frequency bands for each material studied herein.

## 5. Conclusions

The present work proposed a new reproducible acoustic wave generation method for CFRP and aluminum structural monitoring based on launching a metallic sphere and using a piezoelectric diaphragm. The sensor frequency response study conducted by PLB tests showed that the frequency response range for CFRP is about 25 kHz, while for aluminum it is up to approximately 90 kHz. On the other hand, the signal amplitudes observed for CFRP were higher than those for aluminum. Then, a study about the elastic deformations caused by the sphere impact on the surfaces was conducted, in which no damage was observed on the CFRP structure, but there was an apparent mark on the aluminum, which could influence the method repeatability for this material. Subsequently, launch tests were conducted on the CFRP structure at different launch heights, whose results showed high correlation coefficients, revealing the high repeatability of the proposed method, with the highest correlation for the 114 mm height. The results for aluminum launch tests, performed on the launch height and impact distance chosen previously, also produced high correlation indexes between repetitions, demonstrating high repeatability for the method on aluminum and also showing that the elastic deformations caused by the sphere impact were insignificant. It is worth mentioning that the maximum signal amplitude caused by the sphere impact was about ten (10) times higher than those obtained for the PLB in both materials, but the same frequency response ranges were observed. Finally, NDT tests for PLB and launch methods were conducted on both materials, in which signal amplitude variations and frequency shifts were observed with the addition of metallic nuts to simulate damage. Based on the spectra, both CCDM and RMSD indexes showed a progressive increase as the masses were added to the structure, demonstrating the effectiveness of both methods in damage detection for these materials. Also, the magnitude-squared coherence between the launch and PLB signals has shown in most of the useful frequency bands good correlation of both methods. It is worth mentioning that the proposed method presented higher repeatability than the PLB method due to the constant control of the launch conditions (height, sphere mass, launch position, and impact point), which does not happen in the PLB method due to the human inaccuracy in handling the pencil when breaking the lead. Furthermore, the proposed method imposes a much higher acoustic energy source on the structure than the PLB, as verified by the signal magnitudes, which enables a wider reach of the structure. Since a piezoelectric transducer was employed, the launching sphere method is a cheap and simple way to generate an impulsive signal on the structure and a good approach for obtaining the transfer function of the structure being inspected. Future studies to improve the technique can be conducted, such as the evaluation of different diameters and materials for the launch sphere, different structural materials, and real damage detection, such as delamination in CFRP, among others.

## Figures and Tables

**Figure 1 sensors-24-05874-f001:**
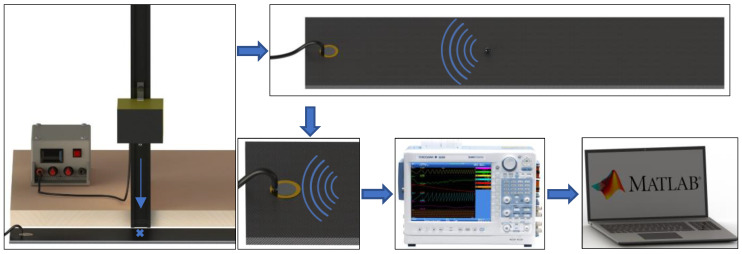
Workbench schematic diagram.

**Figure 2 sensors-24-05874-f002:**
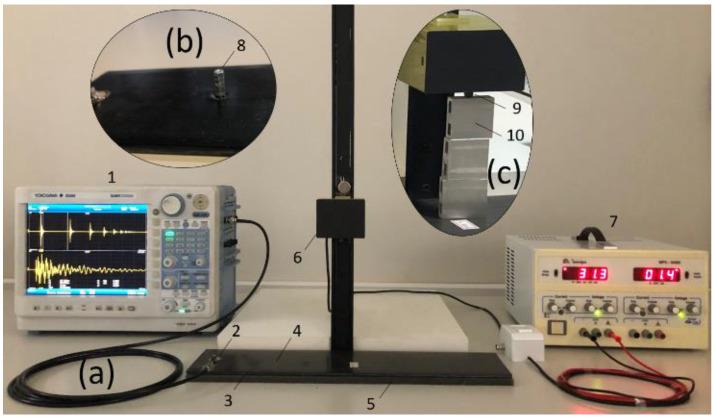
Test bench. In (**a**) oscilloscope (1), piezoelectric diaphragm (2), CFRP or aluminum plate (3), damage addition point (4), foam sheet (5), launching device (6), voltage source (7), metal nuts. In (**b**) (8), metallic sphere. In (**c**) (9), aluminum template or model (10).

**Figure 3 sensors-24-05874-f003:**
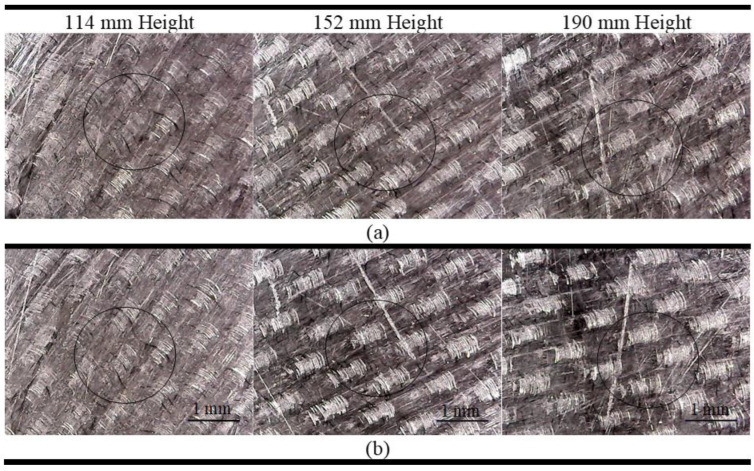
Plastic deformation analysis due to the metallic sphere impact on CFRP: (**a**) before; (**b**) after.

**Figure 4 sensors-24-05874-f004:**
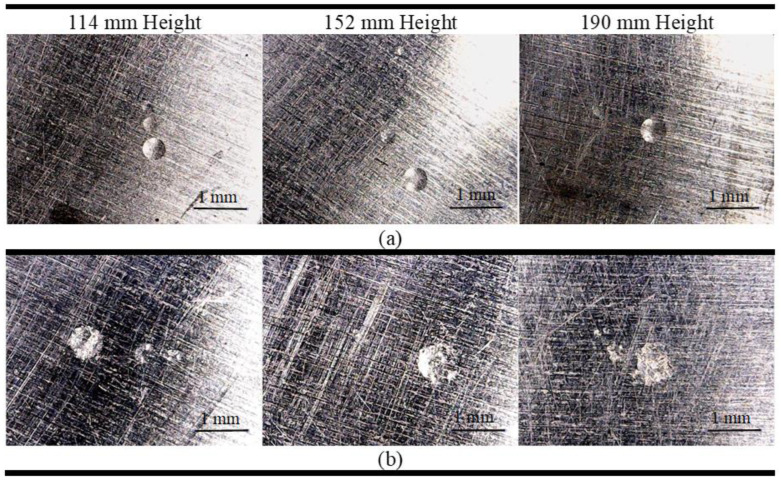
Plastic deformation study due to the metallic sphere impact on aluminum: (**a**) 3.9 mm sphere; (**b**) 6.3 mm sphere.

**Figure 5 sensors-24-05874-f005:**
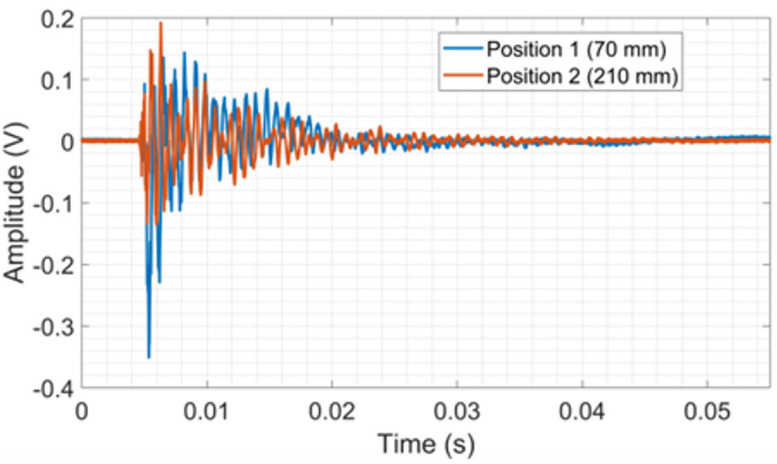
The raw signal of repetition 1 from PLB tests on CFRP.

**Figure 6 sensors-24-05874-f006:**
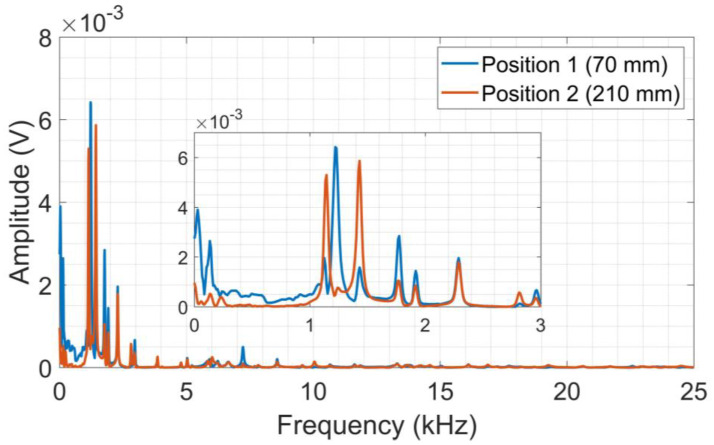
Average frequency spectra from PLB tests on CFRP.

**Figure 7 sensors-24-05874-f007:**
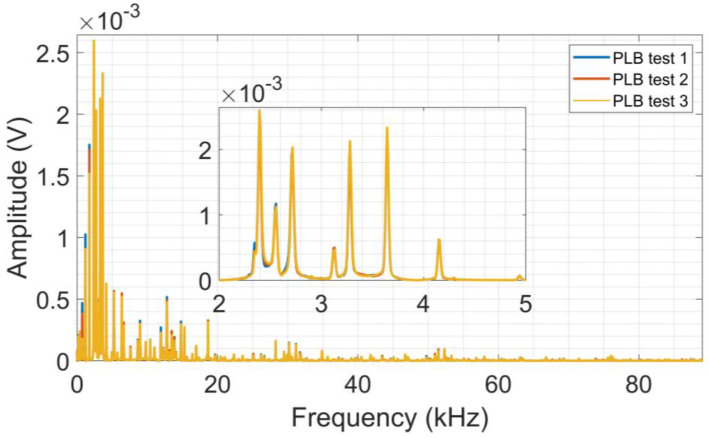
Average frequency spectra from PLB tests on aluminum.

**Figure 8 sensors-24-05874-f008:**
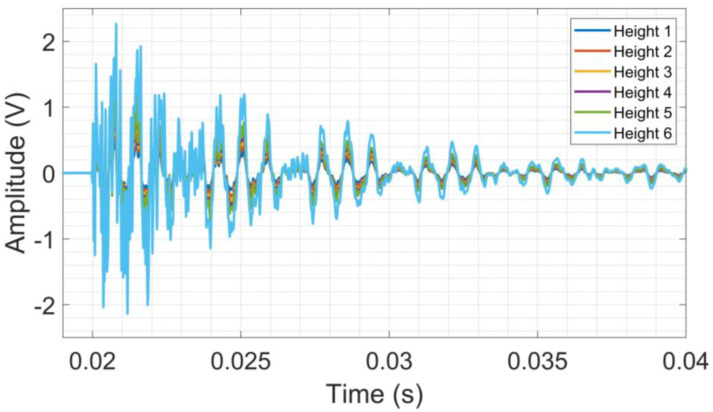
Raw signal from launch tests on CFRP.

**Figure 9 sensors-24-05874-f009:**
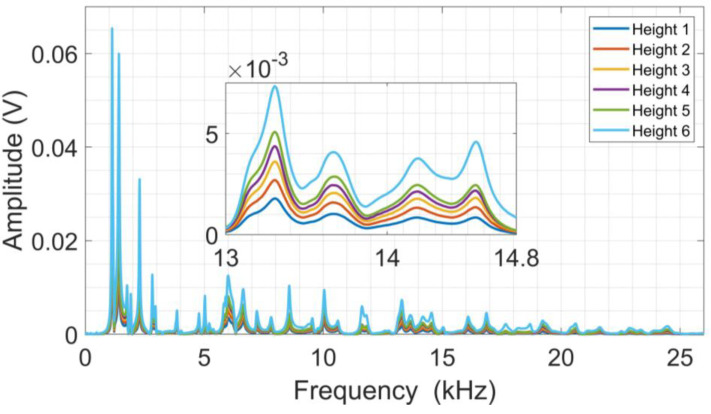
Average frequency spectra from launch tests on CFRP.

**Figure 10 sensors-24-05874-f010:**
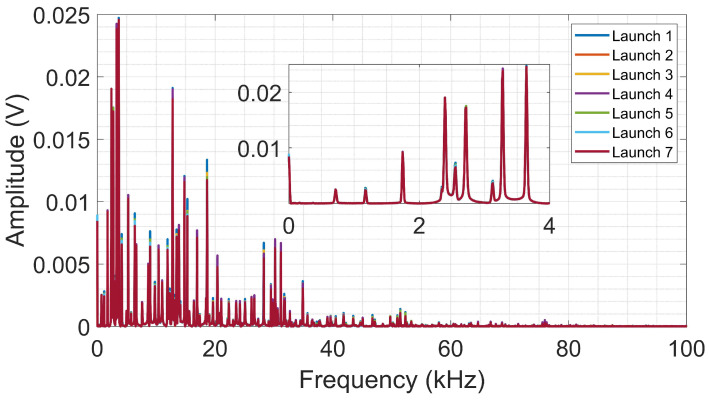
Frequency spectra from launch tests on aluminum.

**Figure 11 sensors-24-05874-f011:**
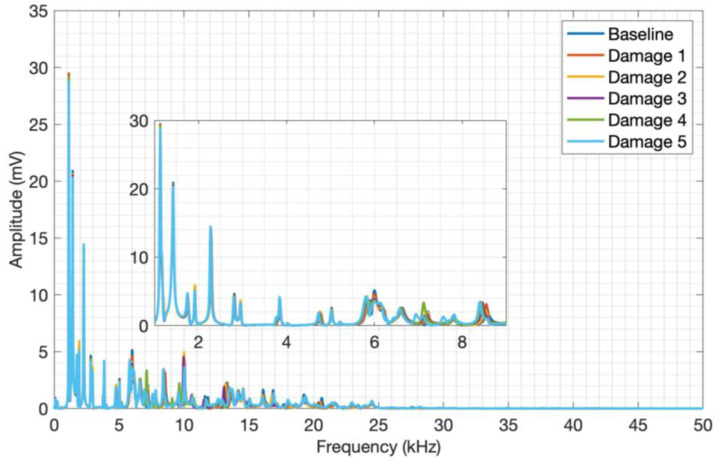
Launch tests adding mass to CFRP.

**Figure 12 sensors-24-05874-f012:**
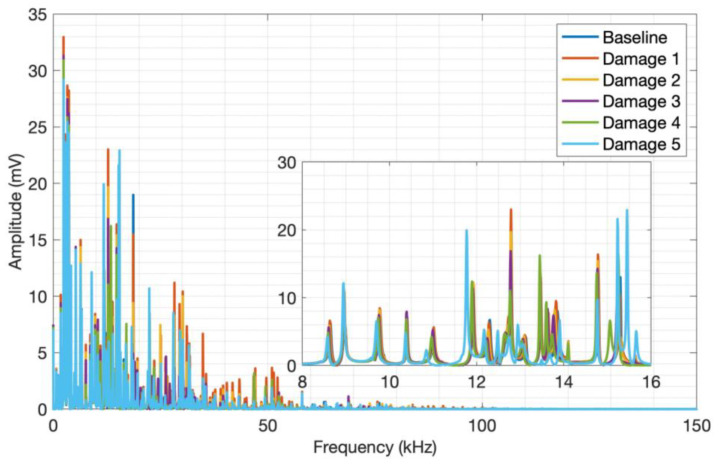
Launch tests adding mass to aluminum.

**Figure 13 sensors-24-05874-f013:**
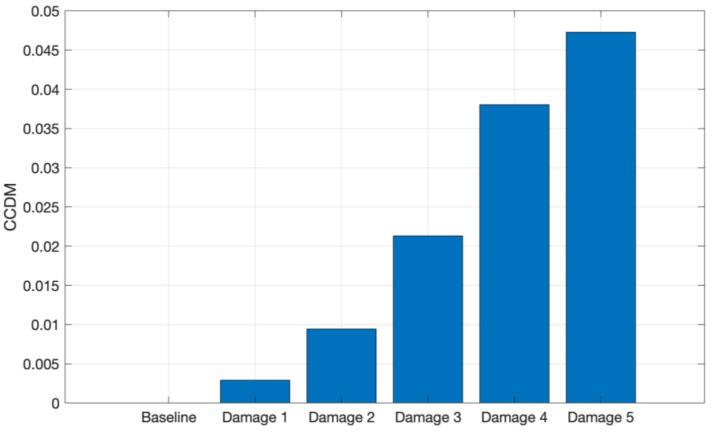
CCDM applied for the tests conducted on CFRP in the frequency band from 1 kHz to 9 kHz.

**Figure 14 sensors-24-05874-f014:**
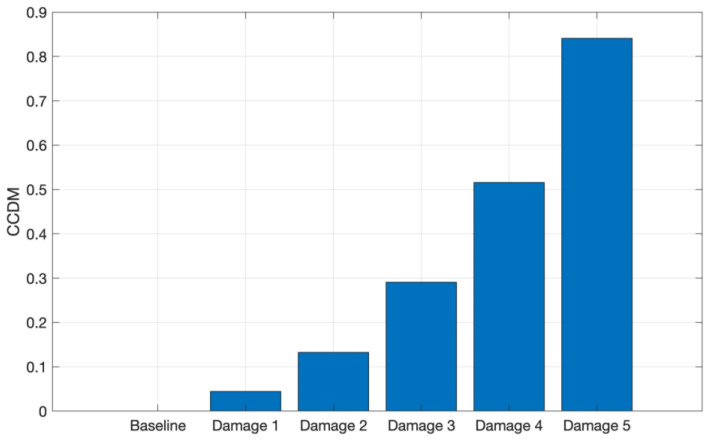
CCDM applied for the tests conducted on aluminum in the frequency band from 8 kHz to 16 kHz.

**Figure 15 sensors-24-05874-f015:**
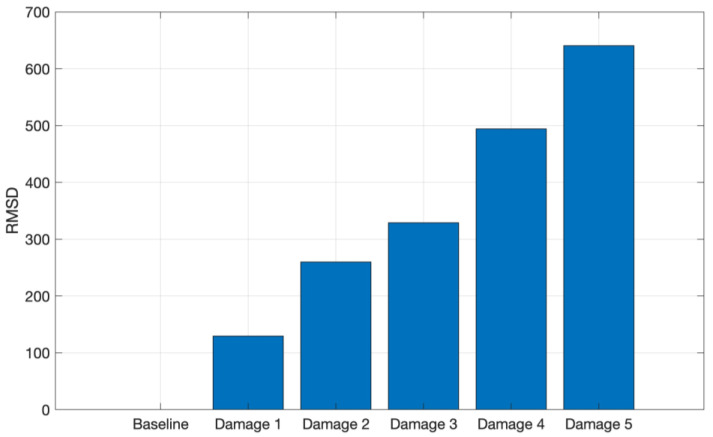
RMSD applied for the tests conducted on CFRP in the frequency band from 8 to 16 kHz.

**Figure 16 sensors-24-05874-f016:**
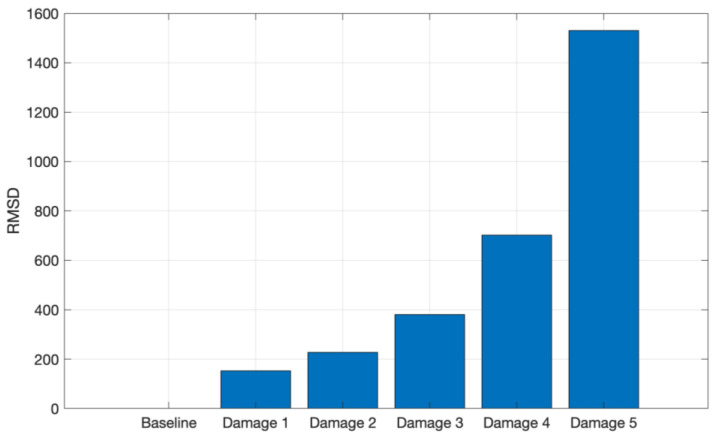
RMSD applied for the tests conducted on aluminum in the frequency band from 8 kHz to 16 kHz.

**Figure 17 sensors-24-05874-f017:**
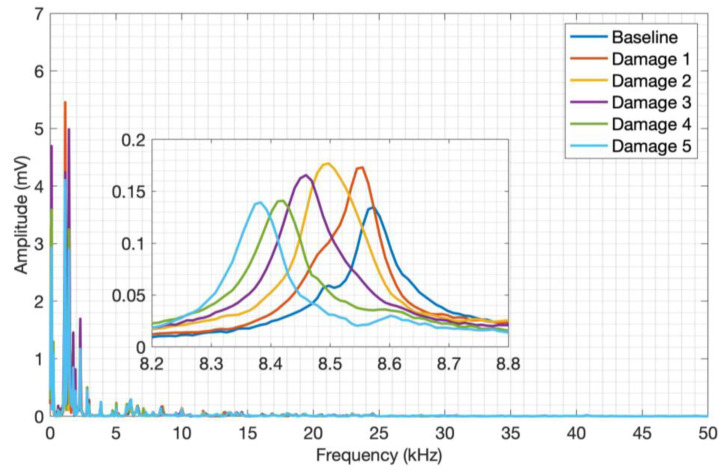
PLB tests adding mass to CFRP.

**Figure 18 sensors-24-05874-f018:**
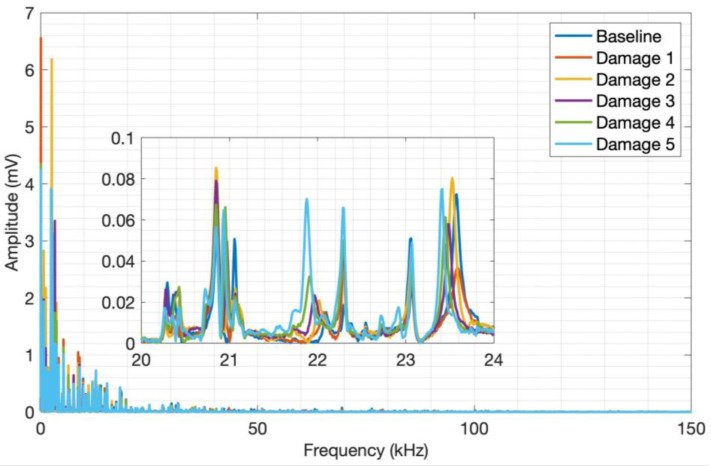
PLB tests adding mass to aluminum.

**Figure 19 sensors-24-05874-f019:**
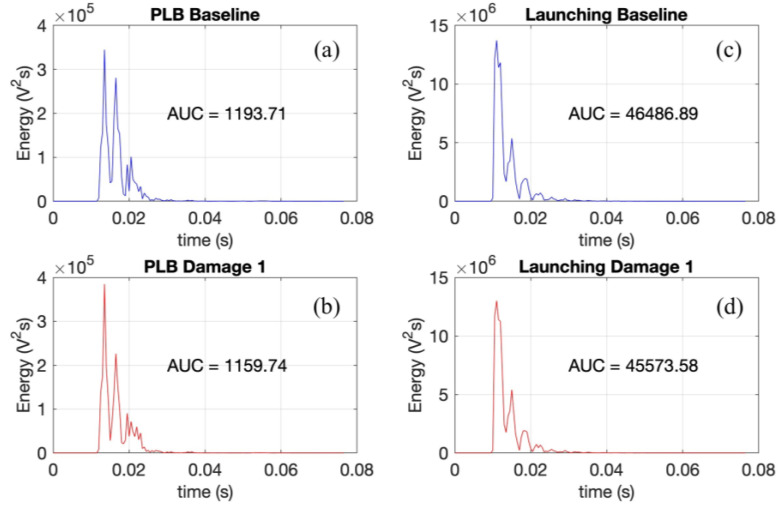
Signal energy for PLB: (**a**) baseline, (**b**) damage 1; Launching: baseline (**c**), (**d**) damage 1.

**Figure 20 sensors-24-05874-f020:**
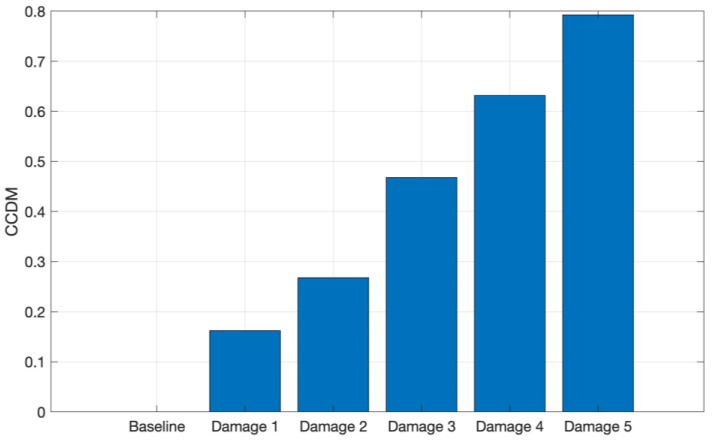
CCDM for CFRP in the frequency band from 20 kHz to 24 kHz.

**Figure 21 sensors-24-05874-f021:**
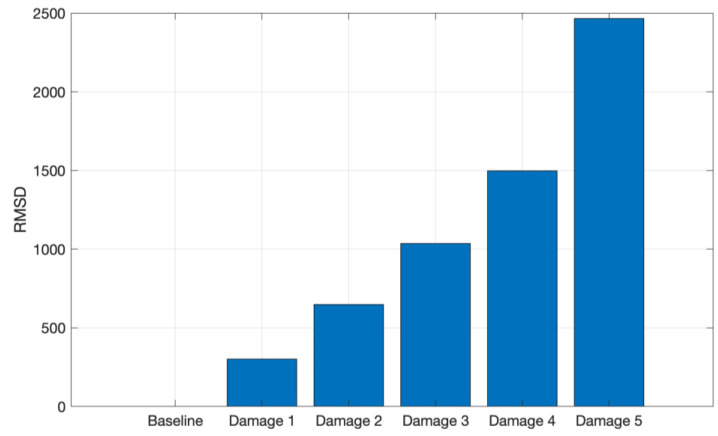
RMSD for CFRP in the frequency band from 20 kHz to 24 kHz.

**Figure 22 sensors-24-05874-f022:**
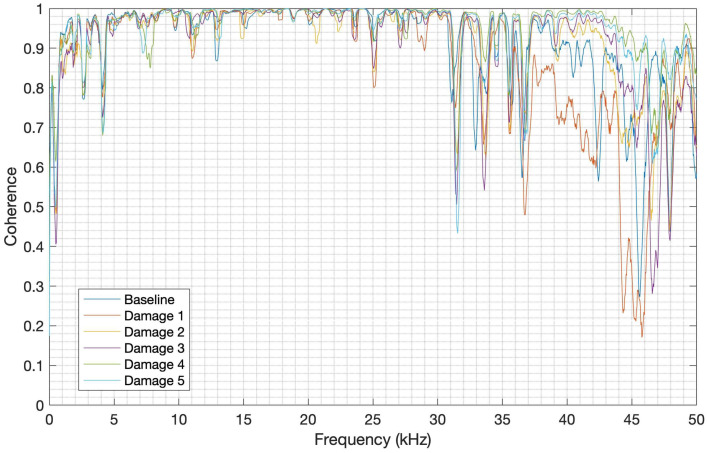
MSC for CFRP.

**Figure 23 sensors-24-05874-f023:**
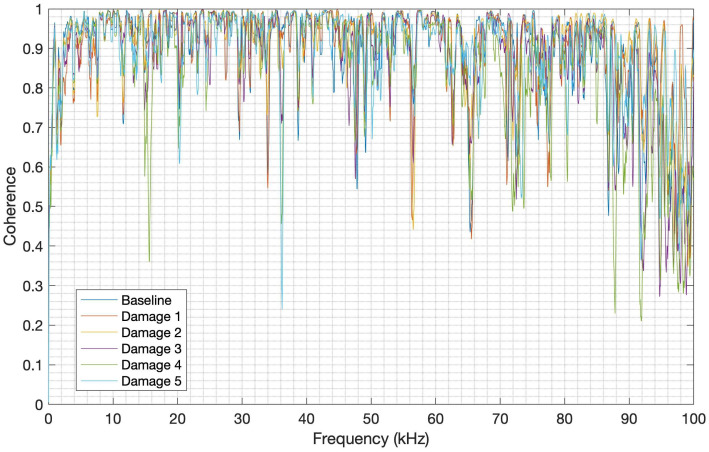
MSC for aluminum.

**Table 1 sensors-24-05874-t001:** Comparison between CFRC and aluminum features.

Feature	CFRC	Aluminum
Density	Variable (around 1.5 g/cm^3^)	Around 2.7 g/cm^3^
Materials	Carbon fibers embedded in a plastic (polymer) matrix	Pure or alloy (copper, magnesium, silicon, or zinc)
Cost	More expansive	Cheaper
Thermal and Electrical Conductivity	Lower	Higher
Strength and Stiffness	Higher	Lower

**Table 2 sensors-24-05874-t002:** Correlation coefficients.

Positions	Average	Standard Deviation
Position 1 (70 mm)	0.9269	0.0480
Position 2 (210 mm)	0.9250	0.0543

**Table 3 sensors-24-05874-t003:** Correlation coefficient for aluminum.

Positions	Correlation Coefficient
Test 1 and test 2	0.9934
Test 1 and test 3	0.9851
Test 2 and test 3	0.9918

**Table 4 sensors-24-05874-t004:** Correlation coefficient for CFRP launch tests.

Launch Height (mm)	Average Correlation Coefficient	Standard Deviation
38 mm	0.9972	0.00240
76 mm	0.9994	0.00014
114 mm	0.9997	0.00003
152 mm	0.9995	0.00026
190 mm	0.9995	0.00027
490 mm	0.9990	0.00073

**Table 5 sensors-24-05874-t005:** Correlation coefficient for aluminum launch tests.

Launches	Correlation Coefficient
Launch 1 and launch 3	0.9945
Launch 1 and launch 5	0.9968
Launch 1 and launch 7	0.9914

## Data Availability

Data is unavailable due to privacy restrictions.

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
