# Peer review of "A Proposed Non-Destructive Method Based on Sphere Launching and Piezoelectric Diaphragm"

_sensors, 2024, doi:10.3390/s24185874_

Round 1

Reviewer 1 Report

Comments and Suggestions for Authors

The authors proposed a new method of SHM of composite and aluminum structures using sphere launching approach and measurement the response with the piezoelectric transducer. The aim of this approach is to establish a repeatable excitation of a structure for reducing possible errors with manual excitation and a classical pencil lead break approach. The idea of the manuscript is original and worth investigation. In the study, the authors compared the results of the proposed excitation method with the pencil lead break method, demonstrating very good accuracy and repeatability of the proposed excitation method. The advantages of the proposed method with respect to the pencil lead break method are clearly demonstrated. The manuscript is well-written and structured. Some minor corrections are, however, required before considering the manuscript for a publication.

  1. The introduction needs to be enriched with an overview of using acoustic emission in SHM of structures, specifically, the methods of excitation, and the goal of the study should be enforced with showing a research gap based on this overview.
  2. In impact evaluation usually the impact energy is provided, please calculate and add information about the impact energy of impacting tested structures by a metallic sphere.
  3. How the position and distance of added mass from the point of impact were selected?
  4. The authors claimed that the testing method is non-destructive, however, the results of microscopic examination show evident dents in aluminum plate and a damage (subsurface delamination) may also appear in composite structures. It is therefore essential to provide parameters at which no damage is introduced due to exciting a tested structure during the test, therefore, the range of heights at which no damage is introduced from one side and contamination of a signal by noise is still acceptable from the other side. This will enhance the applicability of the proposed method.

Reviewer 2 Report

Comments and Suggestions for Authors

This paper introduces a novel non-destructive testing (NDT) method designed to detect damage in carbon fiber-reinforced polymer (CFRP) materials. The proposed approach utilizes a metallic sphere launched onto a piezoelectric diaphragm to generate reproducible acoustic emission signals. The topic is timely and of practical importance. However, the author compares the method proposed in this paper with the PLB method, aiming to demonstrate that the stability of the proposed method is superior to that of the PLB method. Yet, in terms of ease of use, the PLB method is simpler and more practical, and the harm to the object under detection is negligible, making it more suitable for certain scenarios. The application of the method introduced by the author has significant limitations and is not easy to operate in some cases. The author is invited to provide comments on this aspect.

Here are the specific comments:

1.       Consider removing "Study on" from the manuscript title. The term "study" is inherently implied in scientific journal papers.

2.       In Figure 2, please annotate the positions of (b) and (C) with arrows.

3.       There is no period at the end of line 159.

4.       There is garbled text in Table 2.

5.       In line 320, the references [40], [41] should be written as [40, 41].

6.       The conclusion in Figure 9 is self-evident: the higher the height, the greater the potential energy, which corresponds to a larger amplitude.

Comments on the Quality of English Language

There is no problem with the English language.
